# External Validation of Periodontal Screening Using Self-Reports in Dental Settings

**DOI:** 10.3390/diagnostics15233015

**Published:** 2025-11-27

**Authors:** João Viana, Vanessa Machado, Patrícia Lyra, Luís Proença, Naichuan Su, José João Mendes, João Botelho

**Affiliations:** 1Egas Moniz Center for Interdisciplinary Research (CiiEM), Egas Moniz School of Health & Science, 2829-511 Almada, Portugal; vmachado@egasmoniz.edu.pt (V.M.); plyra@egasmoniz.edu.pt (P.L.); lproenca@egasmoniz.edu.pt (L.P.); jmendes@egasmoniz.edu.pt (J.J.M.); jbotelho@egasmoniz.edu.pt (J.B.); 2Department of Oral Public Health, Academic Center for Dentistry Amsterdam (ACTA), University of Amsterdam and Vrije Universiteit Amsterdam, 1081 LA Amsterdam, The Netherlands; n.su@acta.nl

**Keywords:** periodontitis, screening model, self-reported questionnaire, prediction model, CPITN, oral health, model validation

## Abstract

**Background/Objectives:** This study aimed to externally validate the screening model for severe PD proposed by Verhulst in a Portuguese population, assessing its discriminative performance and clinical applicability. **Methods:** A cross-sectional study was conducted with 310 adults recruited from a university dental clinic in Portugal. The participants completed a validated self-reported oral health questionnaire (screening model) and underwent a full-mouth periodontal examination using the CPITN index, with severe PD defined as a score of 4. Model performance was evaluated for discrimination, calibration, sensitivity, specificity, positive predictive value (PPV), and negative predictive value (NPV). **Results:** Of the 310 participants, 51% were diagnosed with severe PD. The Verhulst model showed an area under the curve (AUC) of 0.72 (95% CI: 0.67–0.77), with sensitivity and specificity of 0.709 (95% CI: 0.639–0.779) and 0.724 (95% CI: 0.651–0.796) and PPV and NPV of 0.727 (95% CI: 0.673–0.784) and 0.704 (95% CI: 0.651–0.764), respectively. Calibration was reasonable, with an observed-to-expected ratio of 1.10 (95% CI: 0.939–1.291). **Conclusions:** The model demonstrated acceptable external validity and could serve as a feasible screening tool for severe PD in Portuguese dental settings. Its integration may enhance early diagnosis and interdisciplinary care. Future studies should consider broader diagnostic criteria to refine accuracy.

## 1. Introduction

Periodontitis (PD) is a chronic inflammatory disease of the periodontium that is estimated to affect over 60% of the dentate population worldwide [1]. This disease is characterised by inflammation of the tissues around the tooth, and its screening is a key step to identify people at risk of having the disease undiagnosed [2].

According to the World Health Organization, there is urgency to strengthen the integration of oral health into noncommunicable disease and universal health coverage strategies [3]. This implies that non-dental health professionals should have tools and knowledge to screen periodontitis at primary health care and hospital settings in order to promote an integrated health response [4]. To reach this goal, the Division of Oral Health at the National Centers for Disease Control and Prevention (CDC), in collaboration with the AAP, proposed a self-reported questionnaire [5,6]. This tool has already been validated in other countries worldwide [7,8,9,10,11,12,13,14].

Patients’ self-report is recognized as valuable and reliable information in periodontal medicine with public health benefits [15]. This preventive approach is particularly relevant where an accurate screening process might promote early diagnosis, allowing better therapeutic outcomes and less oral complications [5,6]. Implementing such a strategy in an integrated universal oral health coverage has potential as the restricted human and material resources necessary require approaches that are precise and less time- and labour-intensive [16,17]. Recently, Verhulst et al. [14] validated a screening model for severe PD to be implemented in the medical care setting, with potential for validation in other populations but with the need to adjust individual predicted probabilities [14].

With this in mind, this study aims to validate Verhulst et al.’s screening model for severe PD in a Portuguese population and to examine whether its original and newly computed predicted probability shows adequate performance to be implemented in a medical setting [14].

## 2. Materials and Methods

### 2.1. Source of Data

The present cross-sectional study is reported following the ‘Transparent reporting of a multivariable prediction model for individual prognosis or diagnosis’ (TRIPOD) checklist [18] for prediction model validation. The present study was carried out following the Declaration of Helsinki, updated in 2024, approved by the Egas Moniz Ethical Committee (research number PT-348/23 on 27 February 2024).

In this study, adult subjects (≥18 years) were recruited following consecutive sampling between February 2024 and April 2025 and signed an informed consent. This independent Portuguese cohort was used to externally validate and test the prediction performance of the model originally developed by Verhulst et al. [14], which had been trained and validated in a Dutch population.

### 2.2. Participants

Consecutive enrolment occurred, as adult participants were asked to take part in the initial screening visit at a Portuguese university’s dental clinic (located at the Egas Moniz School of Health & Science). This study’s protocol and objectives were explained to potential participants, who provided written consent prior to involvement. To be eligible, individuals had to be between 18 and 80 years old and possess at least one natural tooth. This criterion was adopted to avoid potential bias arising from the exclusion of individuals who had lost most of their teeth due to severe periodontitis, ensuring that the model accounted for participants across the full spectrum of periodontal health. Including such cases prevented underestimation of disease prevalence and maintained the representativeness of the population, while the CPITN scoring system remained applicable to the existing sextants. Edentulous patients and those who were mentally incapacitated were excluded. Individuals requiring antibiotic prophylaxis before periodontal probing or were unable to communicate in Portuguese were also excluded from participation. Severe periodontitis was defined as a CPITN score of 4, corresponding to the presence of periodontal pockets ≥ 6 mm in at least one sextant. For each participant, the highest sextant score was used as the diagnostic classification, following the CPITN guidelines. To ensure accurate assessment in individuals with reduced dentition, CPITN scoring was applied to all remaining sextants with at least two functional teeth, and the distribution of remaining teeth within this group was recorded. This allowed for consistent and reliable evaluation of periodontal status, even in participants with substantial tooth loss.

### 2.3. Outcome

One calibrated member (J.V.) performed a full-mouth periodontal examination and was blinded for all patient data. Intra-rater reliability was assessed by the intra-class correlation coefficient (ICC) in ten patients not involved in this study, one week apart between each observation. Obtained ICC intra-examiner values were 0.92 and 0.89, for CAL and PD, respectively. Each clinical examination was performed under proper lighting in a dental chair.

All fully erupted teeth, excluding third molars, implants, and retained roots, were examined with manual periodontal measurements (CP-12, Hu-Friedy; Chicago, IL, USA). We recorded the number of missing teeth. Gingival recession (REC), periodontal pocket depth (PPD), and bleeding on probing (BoP) were circumferentially recorded at six sites per tooth (mesiobuccal, buccal, distobuccal, mesiolingual, lingual, and distolingual). PPD, REC, and CAL were assessed as previously detailed [19].

The periodontal chart with full-mouth data was then transformed to community periodontal index of treatment needs (CPITNs) [20]. ‘Severe’ PD was categorized following Verhulst et al. (2019) as a CPITN score of 4 [14].

### 2.4. Predictors

The SROH questionnaire was administered as previously validated [10]. The questions included were those used in the original model developed by Verhulst et al., which identified the most relevant self-reported oral health indicators for predicting severe periodontitis. No additional items were added or excluded. The questions present in this questionnaire had been previously translated and culturally adapted for the Portuguese population following standard cross-cultural validation guidelines to ensure conceptual and linguistic equivalence [10]. It was administered to the participants by one trained researcher (J.V.) using an electronic device linked to a Google Form. The selected questions were as follows: “Q3: Have you ever had treatment for gum disease such as scaling and root planing, sometimes called ‘deep cleaning’?”; “Q4: Have you ever had any teeth become loose on their own, without an injury?”; “Q6: During the past 3 months, have you noticed a tooth that doesn’t look right?”; and “Q8: Aside from brushing your teeth with a toothbrush, in the last 7 days, how many times did you use mouthwash or other dental rinse product that you use to treat dental disease or dental problems?”. Additionally, demographic information (age and sex) was extracted from the comprehensive health and oral hygiene questionnaire completed during the screening process. Demographic and behavioural data were extracted from the screening questionnaire and included age, sex, education level, employment status, and smoking habits. Smoking status was categorized as never smoker, former smoker (individuals who had quit smoking for at least 12 months prior to examination), and current smoker. Employment status was recorded according to self-reported occupation at the time of data collection. Oral hygiene behaviours (e.g., frequency of toothbrushing and mouthwash use) were also documented to complement the self-reported oral health questionnaire.

### 2.5. Sample Size

To estimate the required sample size for validating the model’s performance, we used the reported AUC of 0.75 with a 95% confidence interval of 0.66 to 0.84; the margin of error was calculated as 90%. All analyses were conducted in R version 3.1 (www.r-project.org) using the ‘pROC’ package, we used the Hanley & McNeil (1982) [21] approximation for the standard error (SE) of the AUC, which allows estimation of the sample size required to achieve a specified confidence interval width around an AUC. We assumed a two-sided 95% confidence level (z = 1.96) and solved iteratively for the minimum sample size. Using this approach, we determined that a minimum of 83 cases and 83 controls (total 166 participants) was required to estimate the AUC with the desired precision, at a 1:1 ratio.

### 2.6. Statistical Analysis

After confirming the non-normal distribution of our sample, we performed a Mann–Whitney test for continuous variables and a chi-square test for the categorical variables (*p*-value = 0.00067).

To assess the discriminative performance of the Verhulst model when it is applied in the Portuguese patients, a Receiver Operating Characteristic (ROC) analysis was conducted. The predicted individual risks of severe periodontitis based on the model were compared against the outcome (severe PD) to generate the ROC curve and compute the area under the curve (AUC). An AUC ≥ 0.80 indicates a good to excellent discriminative ability of the model, while an AUC ≥ 0.70 indicates an acceptable discriminative ability. Youden’s Index was calculated, which is the point on the ROC curve closest to the top-left corner, depicting the best performance trade-off (sensitivity and specificity).

To assess the calibration of the model in the Portuguese patients, a calibration plot and the overall observed-to-expected ratios (O:E ratios) were generated [1]. The calibration plot was made by plotting the predicted individual outcomes against the observed actual outcomes [2]. For this, the patients in the dataset were grouped into deciles based on their predicted probabilities for the outcomes. The prevalence of the outcome events within each decile represents the observed probability. The mean of the individual predicted probabilities within each decile represents the predicted probability. In the calibration plot, the observed and predicted probabilities were compared across the range of predicted risk. The O:E ratio was also calculated for the assessment of overall calibration of the models. The O:E ratio was obtained by dividing the prevalence of the outcomes (observed) by the mean of individual predicted probabilities of the outcomes (expected) within the cohort [3]. An O:E ratio < 1 indicates an overestimation of the models, while an O:E ratio > 1 indicates an underestimation of the models [2]. An O:E ratio between 0.8 and 1.2 indicates that the calibration of the model is acceptable [1,3].

In addition, we computed sensitivity, specificity, positive predictive value (PPV), and negative predictive value (NPV) of the models. Sensitivity is defined as the proportion of the patients who had severe PD based on the model among those who had severe PD in reality. Specificity is defined as the proportion of the patients who had no severe PD based on the model among those who had no severe PD in reality. The negative predictive value (NPV) is defined as the proportion of patients without severe PD in reality among those predicted by the model as not having severe PD, while the positive predictive value (PPV) is defined as the proportion of patients with severe PD in reality among those predicted by the model as having severe PD.

All analyses were conducted in R software (version 3.1) (www.r-project.org) and SPSS software 29.0 (IBM, New York, NY, USA). Statistical significance was set at *p* < 0.05.

Schematic diagrams and graphical elements were created with assistance from a generative artificial intelligence tool (ChatGPT (GPT-5.1), OpenAI, San Francisco, CA, USA). No text, data analysis, or interpretation of results was generated using AI tools.

## 3. Results

### 3.1. Participants

From a total of 356 invited participants, 46 were excluded, giving a final sample of 310 patients (Figure 1). The patients had an average age of 52.8 years old (±16.3), with a predominance of 40-year-olds. Among the included patients, 48.0% were males. When asked about their smoking habits, 51.0% of the participants said they were not smokers. Regarding to the employment status and the education level, the majority was employed (59.7%) and had a middle education level (43.9%) (Table 1).

Out of the 310 patients evaluated, 158 (51%) were classified with a CPITN score of 4, while 152 (49%) had scores below 4. The mean age of patients with a CPITN of 4 was 58.5 years (±12.0), whereas the mean age of those with scores below 4 was 46.9 years (±18.0).

### 3.2. Model Performance

The ROC analysis demonstrated an AUC of approximately 0.72 (95% CI: 0.67–0.77) for screening of severe PD (Figure 2). This indicates an acceptable discrimination of the model in Portuguese dental patients. The optimal predicted probability cutoff value was 0.420 based on the ROC curve. This indicates that if the predicted probability of a patient is >0.420, this patient is very likely to have a severe form of PD. If the predicted probability of a patient is <0.420, this patient is unlikely to have severe PD.

The observed-to-expected (O:E) ratio was estimated at 1.10 (95% CI: 0.939–1.291), which indicates that the model had a slight underestimation. That is, the predicted risk was slightly lower than the actual risk of severe PD overall. However, the O:E ratio was between 0.8 and 1.2, which indicates that the calibration of the model was acceptable.

This calibration plot illustrates the agreement between predicted and observed risks (Figure 3). Each point represents a decile of predicted risk, with the *x*-axis indicating the average predicted probability, and the *y*-axis representing the corresponding observed event rate. The 45-degree reference line denotes perfect calibration. In this plot, most points lie close to the diagonal, suggesting acceptable calibration across risk groups. However, slight deviations in some areas may indicate underestimation or overestimation of risk by the model in specific ranges.

Sensitivity was 0.709 (95% CI: 0.639–0.779) and specificity 0.724 (95% CI: 0.651–0.796), with a PPV of 0.727 (95% CI: 0.673–0.784) and an NPV of 0.704 (95% CI: 0.651–0.764).

## 4. Discussion

This prediction validation study aimed to explore the external validity of a model for screening severe periodontitis, proposed by Verhulst et al., in a Portuguese cohort of participants. Overall, the performance of the model was clinically acceptable when it was applied in Portuguese patients in dental settings, with an acceptable discrimination and calibration.

Before screening models can be extensively used in different populations, the performance of the models must be robust across diverse populations to ensure their generalisability and clinical usefulness [22,23]. Therefore, the present study externally validated Verhulst’s prediction model to test this external generalizability and application. In 2019, Verhulst et al. [14] developed three different screening models for severe periodontitis using self-reported oral health, demographics, and/or salivary biomarkers, based on 156 patients from Academic Center for Dentistry Amsterdam (ACTA), The Netherlands. The model showed a good performance for the screening of severe periodontitis, with an AUC of 0.78–0.89. In 2021, Nijland et al. [24] externally validated one of the screening models developed by Martijn, including SROH questions, age, and sex only, based on 159 patients from an outpatient medical setting in The Netherlands. The performance of the model for severe periodontitis in the medical setting was shown to be acceptable, with an AUC of 0.72. In 2025, Nijland et al. [24] further attempted to modify the screening model using five different update methods. However, the updating methods did not further improve the performance of the original screening model for severe PD. Therefore, in the present study, we still used the original model developed by Verhulst et al. for the external validation. Our study was the first one which externally validated the model in the Portuguese population. Despite the expected population-specific factors of this cohort sample (e.g., socioeconomic conditions, access to dental care, and oral hygiene habits), our results provide concrete evidence on the model’s transferability to new settings. Thus, it has the potential to be integrated into routine dental assessments within the Portuguese healthcare context.

The ability of individuals to self-assess their periodontal status reveals two fundamental dimensions [5]. On the one hand, it points to the development of intrapersonal knowledge, reflected in awareness of one’s own oral health [25]. On the other hand, it shows a basic understanding of periodontal disease and its implications, which is essential for encouraging a patient’s active involvement in managing their own health [26]. These assumptions support the importance of health education and individual empowerment, promoting a more autonomous and preventive approach to oral health care [25,27].

The results obtained reinforce the relevance of this approach. The prevalence of severe periodontal disease was significant, with 51% of individuals having a CPITN index of 4 [27,28,29]. This is in line with previous studies, reporting a high prevalence of moderate to severe periodontitis in adult and elderly populations [30,31]. This phenomenon could be explained by the cumulative effect of exposure to risk factors and indicators such as poor oral hygiene, smoking, and systemic diseases, combined with less frequent preventive care in older age groups [32,33]. However, the problem of poor performance of CPITN reported in the literature raises concerns regarding its sensitivity in accurately reflecting the extent and severity of periodontal destruction [10,24,27]. Several studies have highlighted its limitations, including underestimation of disease prevalence, inability to detect clinical attachment loss, and its reduced diagnostic validity in longitudinal monitoring [8,32,33]. Therefore, while the CPITN offers a practical screening tool, caution is warranted in interpreting its findings as representative of true periodontal status [1,8,34,35,36].

In the present study, the discrimination of the model was shown to be clinically acceptable, with an AUC of 0.72. This indicates that if there are 100 pairs of patients, each pair consisting of one patient with the outcome and one without, the model would correctly distinguish 72 of those pairs. The calibration of the model was also clinically acceptable, only with a very slight underestimation. Additionally, the sensitivity (0.709) and specificity (0.724) values reveal a balanced performance of the model in correctly identifying both positive and negative cases. This indicated that the model can be used for both ruling in the severe PD and ruling out the non-severe PD cases. In addition, the positive predictive value (PPV) and the negative predictive value (NPV) are both measures of a model’s performance in clinical decision-making. The PPV represents the proportion of individuals predicted as positive who truly have the condition, while the NPV indicates the proportion of individuals predicted as negative who are indeed free of the condition. Each metric offers complementary insights: the PPV is clinically valuable, as it reflects the likelihood that a positive test result corresponds to a true case, guiding treatment decisions. The NPV, in turn, is essential for ruling out disease, as it provides confidence that a negative test result reliably excludes the condition. In our model, the PPV was 0.727 (95% CI: 0.673–0.784), while the NPV was slightly lower, at 0.704 (95% CI: 0.651–0.764). This also indicated that the model can be reliably used to screen in or out the severe PD, with minimal false positives and false negatives. This shows that the model can be well used in Portuguese dental patients for the screening of severe PD.

Beyond statistical performance, the clinical and societal added value of this screening model should also be emphasised. For clinicians, it provides a simple and efficient tool to identify patients at high risk of severe periodontitis without the need for complex or time-consuming examinations, thereby supporting earlier intervention and personalised preventive strategies. For patients, the model promotes timely diagnosis and treatment, reducing the risk of disease progression, tooth loss, and the associated negative impacts on quality of life. At a societal level, implementing such models may contribute to lowering healthcare costs by reducing the burden of advanced periodontal treatments, decreasing productivity losses linked to oral disease, and ultimately supporting public health strategies aimed at improving population-wide oral health outcomes.

### Strengths and Limitations

This study presents several strengths, including adherence to the TRIPOD checklist, the use of CPITN based on a full-mouth periodontal examination, the employment of a calibrated examiner with high intra-examiner reliability, and a sample size determined through prior calculation, ensuring adequate statistical power. Nevertheless, the use of CPITN as the primary diagnostic criterion represents a limitation, as it may underestimate disease severity and compromise diagnostic accuracy [36]. Future research should consider more comprehensive diagnostic criteria to improve validity.

An important limitation of this study concerns the target population. That is, the patients included in the present study were dental patients. Therefore, the findings are strictly generalisable only to individuals in dental settings, rather than the patients from medical settings. It is essential to acknowledge that medical patients may differ significantly from dental patients in various aspects, such as the prevalence of severe periodontitis, age distribution, and gender proportions. These differences may influence the model’s performance and limit its applicability in broader clinical settings. Moreover, although the model may have considerable potential in medical contexts, where clinical periodontal examinations are more challenging to perform, its performance in such settings was not assessed in the present study. Future research should aim to validate the model in medical populations to ensure its effectiveness and generalisability beyond dental care environments.

## 5. Conclusions

This study validated the screening model by Verhulst et al. in the studied population. The model demonstrated to have an acceptable screening performance, but future research is needed to enhance its clinical utility.

## Figures and Tables

**Figure 1 diagnostics-15-03015-f001:**
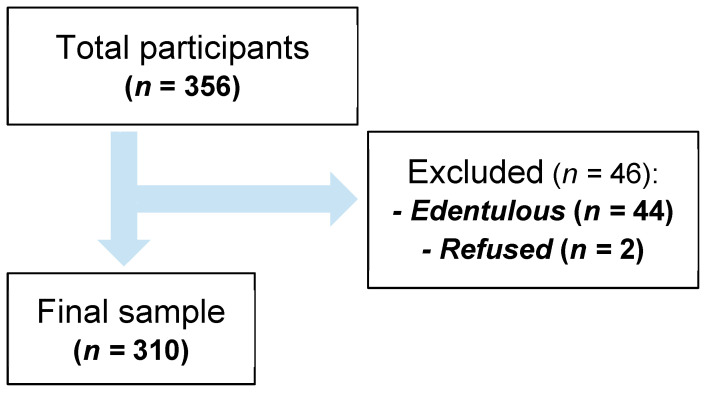
Flow diagram summarising the screening and selection processes.

**Figure 2 diagnostics-15-03015-f002:**
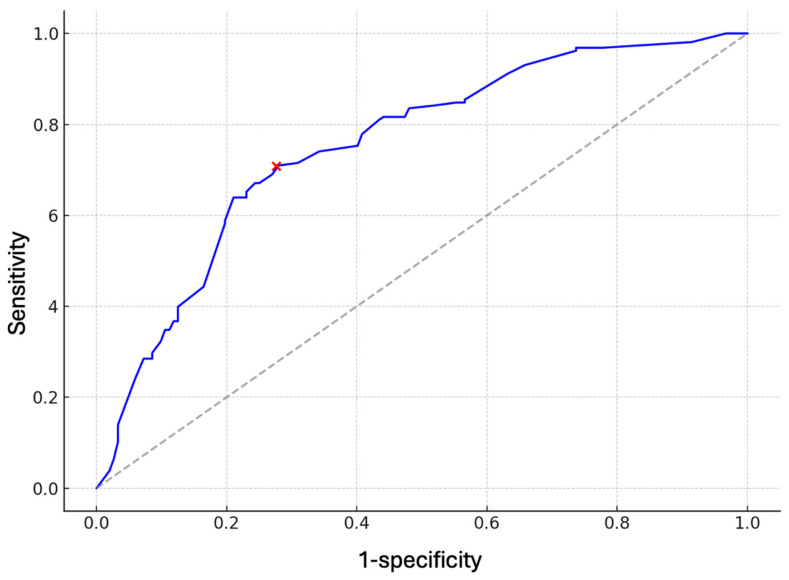
ROC curves with 95% confidence intervals for the screening model for severe periodontitis (the red cross in the curve indicates the optimal cutoff for the predicted risk, which is 0.420).

**Figure 3 diagnostics-15-03015-f003:**
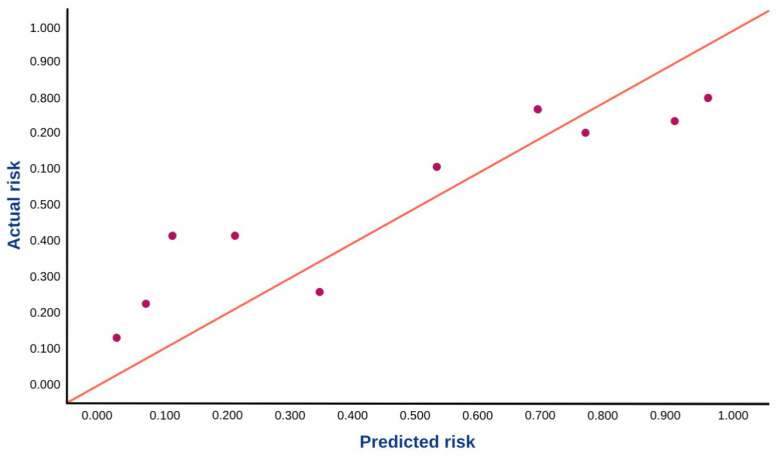
Calibration plot comparing predicted risk with observed risk. The 45-degree line represents perfect calibration.

**Table 1 diagnostics-15-03015-t001:** Participant characteristics according to the periodontal status via CPITN (*n* = 310). Values are presented as mean (standard deviation) for continuous variables and percentage (*n* of participants) for categorical variables.

Variables	Total (*n* = 310)	Severe PD (*n* = 158)	Non-Severe PD (*n* = 152)	*p*-Value
Age, mean (SD) (years)	52.8 (16.3)	58.5 (12.0)	46.9 (18.0)	<0.0001
Sex, % (*n*)				
Female	52.3 (162)	44.9 (71)	59.9 (91)	0.0118
Male	47.7 (148)	55.1 (87)	40.1 (61)	
Smoking, % (*n*)				
Never	51.0 (158)	44.3 (70)	57.9 (88)	0.0552
Former	21.3 (66)	24.7 (39)	17.8 (27)	
Active	27.7 (86)	31.0 (49)	24.3 (37)	
Employment status, % (*n*)				
Student	12.6% (39)	4.4 (7)	21.1 (32)	<0.0001
Employed	59.7% (185)	61.4 (97)	57.9 (88)	
Unemployed	7.4% (23)	8.9 (14)	5.9 (9)	
Retired	20.3% (63)	25.3 (40)	15.1 (23)	
Education, % (*n*)				
Elementary	24.8% (77)	31.6 (50)	17.8 (27)	0.0066
Middle	43.9% (136)	43.7 (69)	44.1 (67)	
Higher	31.3% (97)	24.7 (39)	38.2 (58)	
Number of remaining teeth	20.4 (5.7)	20.5 (5.6)	22.4 (5.6)	0.0007

## Data Availability

The data presented in the study are openly available in 10.5281/zenodo.17711524.

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
