# Peer review of "External Validation of Periodontal Screening Using Self-Reports in Dental Settings"

_diagnostics, 2025, doi:10.3390/diagnostics15233015_

Round 1

Reviewer 1 Report

Comments and Suggestions for Authors

The study is generally well-structured and clearly written. However, a few clarifications and additions would further strengthen the rigor and completeness of the paper. Specific comments are as follows:

1. In section 2.4 (Predictors). Please clarify how the questions included in the SROH questionnaire were selected. Were they based solely on the original Verhulst et al. model, or was there an additional selection or adaptation process for the Portuguese population? A brief justification of the inclusion criteria for these specific questions would improve methodological transparency.

2. Since the diagnosis of severe periodontitis in this study was defined as a CPITN score of 4, it would be helpful to include a description of the specific clinical characteristics and distribution of this group in the methodology section. This will aid readers in understanding the severity profile of the studied population.

3. Inclusion Criterion of “At Least One Natural Tooth”. Please provide a rationale for including participants with only one natural tooth. It would be useful to explain how this criterion may influence the diagnostic reliability of CPITN and whether participants with a minimal number of teeth could potentially bias the screening outcomes.

4. Please add number of Remaining Teeth in Table 1. This parameter is clinically relevant and would provide readers with better insight into the oral health status and potential disease burden of the study cohort.

Author Response

Comment 1#  In section 2.4 (Predictors). Please clarify how the questions included in the SROH questionnaire were selected. Were they based solely on the original Verhulst et al. model, or was there an additional selection or adaptation process for the Portuguese population? A brief justification of the inclusion criteria for these specific questions would improve methodological transparency.

Answer: We thank the reviewer for this valuable comment. The questions included in the SROH questionnaire were the same from the original Verhulst et al. model, but these same questions, that derive from the original AAP SROH, have been previously translated and validated for Portuguese by our group. This same validation was referenced in the methods. We have now clarified in the revised manuscript and this information has been added to Section 2.4 (Predictors).

Comment 2#  Since the diagnosis of severe periodontitis in this study was defined as a CPITN score of 4, it would be helpful to include a description of the specific clinical characteristics and distribution of this group in the methodology section. This will aid readers in understanding the severity profile of the studied population.

Answer: We thank the reviewer for this suggestion. We have now expanded the methodology section to include a clearer description of the clinical characteristics of participants classified as having severe periodontitis (CPITN = 4). In addition, we added information regarding the distribution of remaining teeth and clarified how CPITN scoring was applied in cases with reduced dentition.

Comment 3# Inclusion Criterion of “At Least One Natural Tooth”. Please provide a rationale for including participants with only one natural tooth. It would be useful to explain how this criterion may influence the diagnostic reliability of CPITN and whether participants with a minimal number of teeth could potentially bias the screening outcomes.

Answer: We thank the reviewer for this comment. Edentulous patients were not included because, in the absence of natural teeth, it would not be possible to assess periodontal status or diagnose periodontitis. Including only participants with at least one natural tooth ensures that CPITN scoring can be applied and that periodontal evaluation remains feasible. This also avoids bias by ensuring that individuals who lost most of their teeth due to severe periodontitis are not automatically excluded. We have made adjustments in the Materials and Methods section to clarify this point more explicitly.

Comment 4# Please add number of Remaining Teeth in Table 1. This parameter is clinically relevant and would provide readers with better insight into the oral health status and potential disease burden of the study cohort.

Answer:

We thank the reviewer for this important suggestion. The number of remaining teeth has now been added to Table 1.

Reviewer 2 Report

Comments and Suggestions for Authors

In the methods section, it is stated that “This cohort was used to test the prediction performance of the model developed by Verhulst et al. (18).” This sentence can imply that same data was used in both studies. Please clarify whether the same cohort was also used in the study by Verhulst et al., as this is not entirely clear from the current wording. 

The reference style includes non-English words. Please revise all references to follow an English only format without any non-English terms.

The numbering of the questionnaire items is not sequential (e.g., the first question is labeled Q3, followed by Q4, then Q6 and Q8). Please clarify the total number of questions, indicate whether any questions were omitted from the study, and explain any gaps in numbering.

In addition, please attach the full questionnaire as an appendix to allow complete reporting and to facilitate replication or future studies.

Under the “Predictors” section, please list and define all outcomes that were collected and analyzed so that they correspond with the results section. Some variables mentioned in the results (such as employment or smoking status) are not clearly defined in the methods. For instance, please specify how a “former smoker” was defined (e.g., what is the cut-off period for smoking cessation?). Ensure that all outcomes are reported and clearly defined in the Methods section.

Please provide more information about the dental setting in which participants were enrolled. Indicate whether it was a general dental clinic, a screening center, or a specialized setting such as a periodontal clinic.

The Statistical Methods section does not specify how the two groups were compared. Please report the statistical tests used for between-group comparisons, both in the Methods section and in the footnote of Table 1.

Minor Comments

Please standardize author abbreviations throughout the manuscript. For example, change “JV” to “J.V.” for consistency.

In Table 1, the percentage of males with non-severe periodontitis is missing. Please add this information for completeness.

Author Response

Comment 1#: In the methods section, it is stated that “This cohort was used to test the prediction performance of the model developed by Verhulst et al. (18).” This sentence can imply that same data was used in both studies. Please clarify whether the same cohort was also used in the study by Verhulst et al., as this is not entirely clear from the current wording.

Answer: We thank the reviewer for this observation. We confirm that the cohort used in the present study is independent from the one used by Verhulst et al. The current sample comprises Portuguese adults who attended the Egas Moniz Dental Clinic, whereas Verhulst et al. developed and validated their model using a Dutch population-based cohort. Our study aimed to externally validate their predictive model in a different demographic and clinical context, thereby assessing its generalizability and performance across populations.

Comment 2#: The reference style includes non-English words. Please revise all references to follow an English only format without any non-English terms.

Answer: We are thankful for pointing this out. We apologize for the typos that have been corrected accordingly.

Comment 3# The numbering of the questionnaire items is not sequential (e.g., the first question is labelled Q3, followed by Q4, then Q6 and Q8). Please clarify the total number of questions, indicate whether any questions were omitted from the study, and explain any gaps in numbering.

Answer: We thank the reviewer for this pertinent observation. The SROH questionnaire originally includes eight items (Q1–Q8). In the present study, only four questions (Q3, Q4, Q6, and Q8) were selected, as these correspond exactly to the self-reported oral health items included in the predictive model developed by Verhulst et al. (reference 10). The remaining questions were excluded because they were not part of the original model’s predictor set and were therefore not relevant to the external validation process. The original numbering of the items was retained to maintain consistency with the validated SROH structure and with the previous publications by Verhulst et al.

Comment 4#: In addition, please attach the full questionnaire as an appendix to allow complete reporting and to facilitate replication or future studies.

Answer: We thank the reviewer for this helpful suggestion. The full version of the SROH questionnaire, including all eight items (Q1–Q8) and their respective English and Portuguese versions, has now been added as Appendix to ensure methodological transparency and facilitate replication in future studies.

Comment 5#: Under the “Predictors” section, please list and define all outcomes that were collected and analysed so that they correspond with the results section. Some variables mentioned in the results (such as employment or smoking status) are not clearly defined in the methods. For instance, please specify how a “former smoker” was defined (e.g., what is the cut-off period for smoking cessation?). Ensure that all outcomes are reported and clearly defined in the Methods section.

Answer: We thank the reviewer for this valuable observation. We have now expanded the “Predictors” section to include a detailed description of all variables collected and analysed, ensuring full alignment with the Results section. Specifically, demographic (age, sex, education level, employment status) and behavioural variables (smoking and oral hygiene habits) are now explicitly defined. Smoking status was categorized as never smoker, former smoker, defined as an individual who had stopped smoking for at least 12 months prior to the examination, and current smoker. Employment status was defined according to self-reported occupation at the time of data collection. These clarifications have been added to improve methodological transparency and reproducibility.

Comment 6# Please provide more information about the dental setting in which participants were enrolled. Indicate whether it was a general dental clinic, a screening center, or a specialized setting such as a periodontal clinic.

Answer: We appreciate the reviewer’s comment. This information is already described in the Materials and Methods section, where it is stated that data were collected at the Egas Moniz Dental Clinic, a university-based general dental clinic that provides comprehensive dental care and routine oral health screenings to the community. To improve clarity, we have now slightly reworded the sentence to explicitly indicate the nature of this clinical setting.

Comment 7# The Statistical Methods section does not specify how the two groups were compared. Please report the statistical tests used for between-group comparisons, both in the Methods section and in the footnote of Table 1.

Answer: We apologize for missing this information. Due to the fact this sample presents a non-normal distribution, we performed non-parametric tests. We performed Mann-Whitney test for continuous variables and chi-square test for categorical variables. We added this information as follows in section 2.6: “After confirming the non-normal distribution of our sample, we performed Mann-Whitney test for continuous variables and chi-square test for categorical variables.”

Minor comment 1# Please standardize author abbreviations throughout the manuscript. For example, change “JV” to “J.V.” for consistency.

Answer: We thank the reviewer for noticing this detail. All author abbreviations have now been reviewed and standardized throughout the manuscript to ensure consistency.

Minor comment 2# In Table 1, the percentage of males with non-severe periodontitis is missing. Please add this information for completeness.

Anwer: We thank the reviewer for pointing this out. The percentage of males with non-severe periodontitis has now been added to Table 1 to ensure completeness and consistency with the other reported categories. The table and corresponding values have been carefully checked for accuracy.

Round 2

Reviewer 1 Report

Comments and Suggestions for Authors

The revised version shows improvement in clarity and overall presentation. As a minor revision, kindly add the p-value for the number of remaining teeth in Table 1.

Author Response

"The revised version shows improvement in clarity and overall presentation. As a minor revision, kindly add the p-value for the number of remaining teeth in Table 1."

Our answer: Dear reviewer, we appreciate the carefulness. We apologize for the typo. We added the respective p-value of 0.0007 in Table 1. This change may be found respectively in the mentioned table of the revised manuscript.